# Experimental Investigation of the Mechanical Properties and Fire Behavior of Epoxy Composites Reinforced by Fabrics and Powder Fillers

Kamila Sałasińska [1,*], Mikelis Kirpluks [2], Peteris Cabulis [3], Andrejs Kovalovs [3], Eduard Skukis [3], Paweł Kozikowski [1], Maciej Celiński [1], Kamila Mizera [1,4], Marta Gałecka [4], Kaspars Kalnins [3] and Ugis Cabulis [2]

[1] Department of Chemical, Biological and Aerosol Hazards, Central Institute for Labour Protection—National Research Institute, 16 Czerniakowska St., 00-701 Warsaw, Poland; pawel.kozikowski@ciop.pl (P.K.); maciej.celinski@ciop.pl (M.C.); kamila.mizera@ciop.pl (K.M.)

[2] Polymer Laboratory, Latvian State Institute of Wood Chemistry, 27 Dzerbenes St., LV-1006 Riga, Latvia; mikelis.kirpluks@kki.lv (M.K.); cabulis@edi.lv (U.C.)

[3] Institute of Materials and Structures, Riga Technical University, 6b Kipsalas St., LV-1048 Riga, Latvia; peteris@ritols.lv (P.C.); andrejs.kovalovs@latnet.lv (A.K.); edskukis@gmail.com (E.S.); kaspars.kalnins@latnet.lv (K.K.)

[4] Faculty of Materials Science and Engineering, Warsaw University of Technology, 02-507 Warsaw, Poland; sgalecki44@gmail.com

\* Correspondence: kamila.salasinska@ciop.pl

**Abstract:** Different types of fabrics, such as aramid (A), carbon (C), basalt (B), glass (G), and flax (F), as well as powder fillers, were used to manufacture the epoxy-based hybrid composites by the hand-lay-up method. In this work, a few research methods, including hardness, flexural tests, puncture impact behavior, as well as cone calorimetry (CC) measurements, were applied to determine the impact of type fillers and order of fabrics on the performance and burning behavior of hybrid composites. The mechanical properties were evaluated to correlate with the microstructure and consider together with thermogravimetric analysis (TGA) data.

**Keywords:** glass fabrics; aramid fabrics; carbon fabrics; basalt fabrics; flax fabrics; vermiculite; glass spheres; hazelnut shell

## 1. Introduction

Polymer hybrid composites are materials with a plastics matrix, in which various types of reinforcement, such as glass, carbon, aramid, thermoplastic, basalt or natural (linen, hemp, sisal) fibers, are combined. In a hybrid composite, subsequent layers are made of various fabrics, either composite is made of hybrid fabrics with a weft made of various fibers. A properly designed and made composite has individual components' advantages while compensating for others' disadvantages. For example, to obtain a composite with excellent mechanical properties while reducing the product's price some of the more expensive fibers are replaced with their cheaper counterparts [1]. A promising direction in hybrid composites production is the combination of various types of continuous fibers with powder fillers [2].

The possibility of modifying composite materials' functional properties in a wide range meant that they found their application in various industries. They are perfect as materials for the construction elements of vehicles, aircraft fuselages or in the production of battle covers. Moreover, they are more and more often used in medicine, e.g., for the production of prostheses. Systems of organofunctional alkoxides and trialkoxysilanes on supports, inter alia, made of epoxy resin reinforced with glass fiber, are used to produce microlenses, optical connectors, and waveguides [3–6].

The composites' performance depends mainly on reinforcing fibers' properties, its amount, the saturation by matrix, interfacial adhesion, as well as the structure of the

fabric and structural defects resulting from the manufacturing process. According to the literature, the density, tensile strength, as well as Young's modulus depending on the type and producer are as follows: Glass fibers 2.49–2.7 g/cm$^3$, 2800–3600 MPa and 70–96 GPa, carbon fibers 1.5–1.9 g/cm$^3$, 900–4500 MPa, 40–460 GPa, polyamide fibers 1.39–1.45 g/cm$^3$, 700–3620 MPa, 17–129.4, basalt fiber 2.75 g/cm$^3$, 3840 MPa, 89 GPa, natural fibers 0.7–1.54 g/cm$^3$, 442–2000 MPa, 17–70 GPa [3,7–9]. There are many publications [9–12], discussing the change of properties conditioned by such aspects as the type of reinforcement, the number of layers, and the order of fabrics, as well as the direction of the fibers.

The article [13], written by Zachariah et al., presented hybrid carbon/aramid epoxy composites with aramid layers at different layup sequences, subjected to impact and flexural loading. The authors observed that the hybridization of the carbon and aramid fabrics have a positive effect on impact strength. Furthermore, it was proved that the sequence and the position of aramid fabrics exhibit an important role in enhancing the impact performance. The investigation of the influence of the matrix properties, the number of the carbon and aramid layers, as well as fiber orientation on low-velocity impact behavior of the hybrid epoxy composites, is presented in the article [14]. The composite material with a greater number of the carbon fabrics with fiber orientation at 0° to the load direction exhibited the best impact response and the lesser damage degree. Moreover, it was determined that the composite materials' energy absorption capacity depends on the number of carbon and aramid layers. The differences in the properties of the composite with the epoxy resin matrix reinforced with glass and aramid fibers, as well as glass and carbon fibers are presented in the article [4]. Replacing aramid fibers with carbon fibers allowed more than a 2-fold increase in the tensile strength and Young's modulus. Moreover, better results for flexural strength were achieved by loading the composite from the side of glass fabrics. In turn, at 50 °C a slightly higher value of the modulus was obtained by loading the material from the aramid side, while in the case of the combination of glass and carbon fibers, the load side was irrelevant [4]. In turn, in the article [15], the carbon fabric epoxy composites (CF/EP) reinforced by graphene oxide-coated short glass fibers (GO/SGF) were manufactured and characterized. The results show that bonding via hydrogen bonding with GO improved the compatibility of the SGF in the epoxy matrix. Moreover, the addition of GO has greatly increased the surface roughness of SGF, which was conducive to mechanical interlocking with the EP matrix. The shear strength of the composite reached the maximum value at the content of GO/SGF equal to 0.4 wt%. Matykiewicz and Barczewski [16] evaluated the thermomechanical properties of hybrid epoxy composites reinforced with flax and basalt fiber modified with silanized basalt powder. It turned out that incorporating flax fibers as an internal reinforcement layer does not negatively affect the epoxy composites' mechanical properties. Moreover, the introduction of silanized basalt into the epoxy matrix improved the composites' stiffness in flexural tests, and the highest value was demonstrated by samples containing only 5 wt%.

When selecting the reinforcement, many properties should be taken into account, such as the fibers' strength (carbon fillers), chemical and thermal resistance (glass and basalt fillers) or density (polymer and plant fillers). The features contribute to the product's final properties, but also the manufacturing process and price. Recently, more attention is paid to the availability and sustainability of fillers that make up the final product more environmentally friendly. The energy consumption of the acquisition and production process is not without significance, as well as the possibilities to use post-production waste from the agricultural or mining industry. The motivation of the research was to combine the different fillers to improve the performance of epoxy composites. For this purpose, in addition to high-performance fabrics providing fire protection and good mechanical properties, plant fillers and powder substitutes, which reduced the costs and carbon footprint of a product, were used. In this work, a few research methods, such as hardness, flexural tests, puncture impact behavior, as well as cone calorimetry measurements, were

used to define the influence of type of fillers and order of fabrics on the performance and burning behavior of EP composites.

## 2. Materials and Methods

The epoxy resin, based on bisphenol A diglycidyl ether, RenLam LY 113 (viscosity 580 MPa·s at 25 °C, density 1.16 g/cm$^3$) and a hardener Ren HY 97-1 (viscosity 20 MPa·s at 25 °C, density 0.95 g/cm$^3$), both from Huntsman Advanced Materials GmbH (Basel, Switzerland), were used as the polymer matrix. Processing of the composites was accomplished using the same resin-to-curing agent ratio of 100:30 by weight.

In turn, five kinds of fabrics were used as reinforcement, i.e.,: a two-way (+45/−45°) sewn glass X-E (G) made of E-glass and with a grammage of 444 g/m$^2$ from Saertex GmbH & Co. KG (Saerbeck, Germany), a two-way (+45°/−45°) sewn carbon fabric X-C (C) with a grammage of 406 g/m$^2$ from Saertex GmbH & Co. KG (Saerbeck, Germany), a two-way (+45°/−45°) sewn basalt fabric BAS BI 450 (B) with a grammage of 464 g/m$^2$ from Basaltex NV (Wevelgem, Belgium), aramid fabric (A) made of Tex 240 fibers with a 2 × 2 twill weave and a weight of 300 g/m$^2$ delivered by P.P.H.U. SURFPOL (Rawa Mazowiecka, Poland), as well as a flax fabric (F) with a 2 × 2 twill weave and a grammage of 500 g/m$^2$ made of Tex 400 fibers manufactured by Safilin (Sailly-Sur-La-Lys, France).

Raw vermiculite (V) with a grain size in the range of 0.3–1 mm, and bulk density of 110–130 kg/m$^3$ were purchased from Vermeko Ltd. (Lublin, Poland). V is composed mainly of $SiO_2$, $MgO$, $Al_2O_3$, and $FeO_3$, and the admixture of other rocks does not exceed 10%. Moreover, the glass spheres MIKROBALON DT-99 (M) delivered by PROGMAR (Leszno, Poland) filled with $CO_2$, a diameter of 30–115 μm and a weight of about 120 g/L, were applied. Hazelnut shell (S) obtained from AGRO Jarosław Seroczyński (Nadbrzeż, Poland) were used as a plant-derived filler. The preparatory procedure included drying and grinding with the use of MUKF-10 laboratory sieve mill (sieve mesh size of 0.2 mm) from Młynpol (Chwaszczyno, Poland).

The traditional manual hand-lay-up method was employed to manufacture the hybrid composites in the present study. Each composite consisted of five two-plies of layers, and sometimes one of the plies has been replaced by a powder filler, as indicated in Table 1. Firstly, the fabrics were cut into pieces with dimensions of 330 × 330 mm. Then, epoxy resin was prepared by mechanical mixing of the components, using a proLAB 075 from GlobimiX ( Ząbkowice Śląskie, Poland) stirrer with a rotational speed of 2000 rpm for 3 min and under subatmospheric pressure. The fiber layers were positioned one after one on the polyethene plates, and in the meantime, an epoxy resin was uniformly dispersed using a roller. For each kind of laminates a similar amount (520 ± 25 g) of the epoxy composition were used. After the forming process, the composites were cured at room temperature for 72 h and post-cured for the next 3 h at 70 °C using a Goldbrunn 1450 vacuum dryer from Goldbrunn Therm (Zielona Góra, Poland). Finally, the laminates were cut neatly as required standards for performing various experiments using jigsaw KAPEX KS 120 REB from Festool (Wendlingen, Germany). All the obtained laminates had a thickness equal to 6.4 ± 0.5 mm.

**Table 1.** Laminate stacking sequences.

| Samples | G | C | A | B | F | V | M | S |
|---|---|---|---|---|---|---|---|---|
| GCABF | 2a | 2b | 2c | 2d | 2e | | | |
| CGABF | 2b | 2a | 2c | 2d | 2e | | | |
| CAGBF | 2c | 2a | 2b | 2d | 2e | | | |
| MAGCBF | 2b | 2c | 1a | 2d | 2e | | 1a | |
| VGACBF | 1a | 2c | 2b | 2d | 2e | 1a | | |
| MGACBSF | 1a | 2c | 2b | 2d | 1e | | 1a | 1e |

1, 2 is a number of layers, while a–e is an order.

Cross-sections of the obtained composites were examined using an ultra-high resolution scanning electron microscope SU8010 from Hitachi (Tokyo, Japan). To improve conductivity, the material was gold-coated using a Quorum Technologies Q150T ES from Quorum (Laughton, UK) sputter coater. Observations were conducted in the secondary electron (SE) mode, at an accelerating voltage of 10 kV, at magnification ×100 and the biggest possible working distance, usually WD > 30 mm, to maximize field depth and minimize image distortion. Each sample was properly oriented and observed from top to bottom. Images were taken at a constant stage shift to unsure a sufficient area overlap. For each sample, partially overlapping images were stitched together using Grid/Collection Stitching plugins available in the open-source image processing package Fiji suite [17,18]. The number of images depended on the composite height, usually, each consisted of 8–10 images. The stitched images' final resolution was around 2600 × 13,000 pixels, which corresponds to a horizontal field of view of 1.3 mm and a vertical field of view of >6 mm.

The hardness of composites was measured using a Durometer D from Wilson Wolpert Company (Fort Worth, TX, USA), according to DIN 53505, ASTM D2240, and ISO 7619.

The flexural tests were executed using a three-point bending technique according to the ISO 14,125 standard with an ElectroPuls E3000 from the Instron (Norwood, MA, USA) testing machine at room temperature. The measurements were conducted on specimens with 6 × 15 mm dimensions at a crosshead speed of 7 mm/min and with a support spacing of 110 mm.

The low-speed impact test was performed on the INSTRON Dynatup 9250 HV Impact Tower test machine (Norwood, MA, USA). The total mass of drop weight framework with a hemispherical striker equals 5.83 kg. The hemispherical striker with a diameter of 25.4 mm was connected with a force transducer of 16 kN capacity to measure the impact force during the test. The square specimen (100 × 100 × 6 mm) was clamped between two steel plates with a cut circular hole of diameter 40 mm. The falling height between the striker and top surface of the specimen is 7.0 m. This fall height was achieved by the spring prestress in the test machine. The impact energy can be calculated from the equation of potential energy using a mass of drop weight framework and falling height. In the present case, the impact energy of 400 J was calculated. The calculated impact energy was chosen for the failure of all samples without a rebound. During an experiment absorbed energy, velocity and deflection were calculated using the in-built software. Three specimens of each composition were tested.

The thermogravimetric analysis (TGA) was performed on a TGA Q500 analyzer from TA Instruments (New Castle, DE, USA) at a heating rate of 10 °C/min. The samples (~20 mg) were tested under flowing nitrogen, at a rate of 10 mL/min in the chamber and 90 mL/min in the oven, over a temperature range from ambient temperature to 900 °C.

Fire behavior was assessed using a cone calorimeter from the Fire Testing Technology (East Grinstead, UK), following the ISO 5660 standard procedure. The square samples (100 × 100 × 6 mm) were tested at an applied heat flux horizontally of 35 kW/m$^2$. The residues were photographed using a digital camera EOS 400 D from Canon Inc. (Tokyo, Japonia). Moreover, the samples after cone calorimetry were also investigated by SEM. Each layer was delaminated by tweezers and did not require sputter coating. Due to their high porosity and brittleness, composite chars have been fixed on the tape using LEIT-C Conductive Carbon Cement. Both top surface and bottom chars have been observed and analyzed, as well as each fiber type.

## 3. Results

### 3.1. Surface Morphology Analysis

The composites' morphology was assessed using a scanning electron microscope, and the obtained images of the sample cross-sections are summarized in Figure 1. Observations confirmed that the use of a low-viscosity resin had a positive effect on fabrics' saturation. Nevertheless, the images also showing bundles of both stiff fibers from sewn fabrics (glass, basalt, carbon) and tangled fibers from woven fabrics (aramid, linen) with a smooth surface,

prove the limited adhesion of resin and reinforcements. The mixing process resulted in a loss of structure in spheres, and the images show the agglomerates of the glass powder. Agglomerates were especially visible in the case of vermiculite, while not seen for the ground hazelnut shell. Furthermore, a few voids could be seen in certain areas due to the matrix's poor saturation, which is characteristic of the hand-lay-up method. Another disadvantage of this method is the significant amount of resin and the matrix's unfavorable ratio to reinforcement. The good laminates' properties are dependent on the existence of a lesser number of voids, proper breakage with the fiber pull out and better adhesion behavior between the fibers and matrix [19].

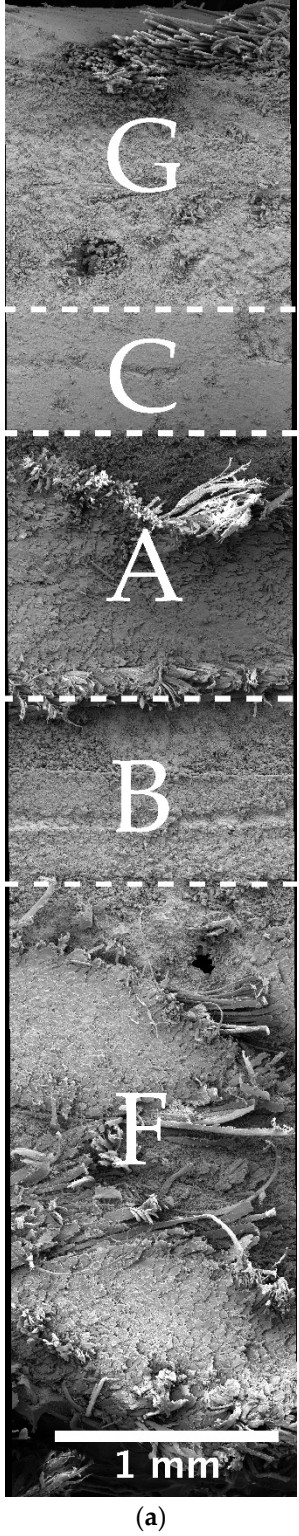

(**a**)

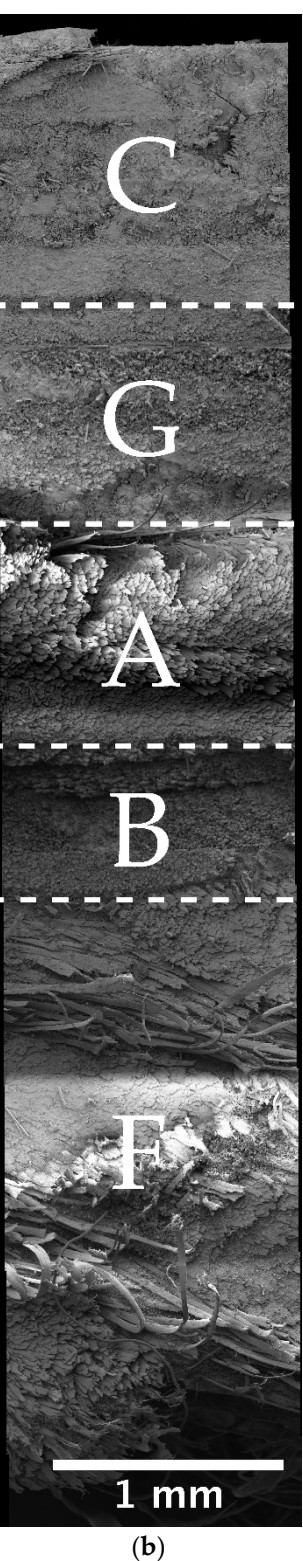

(**b**)

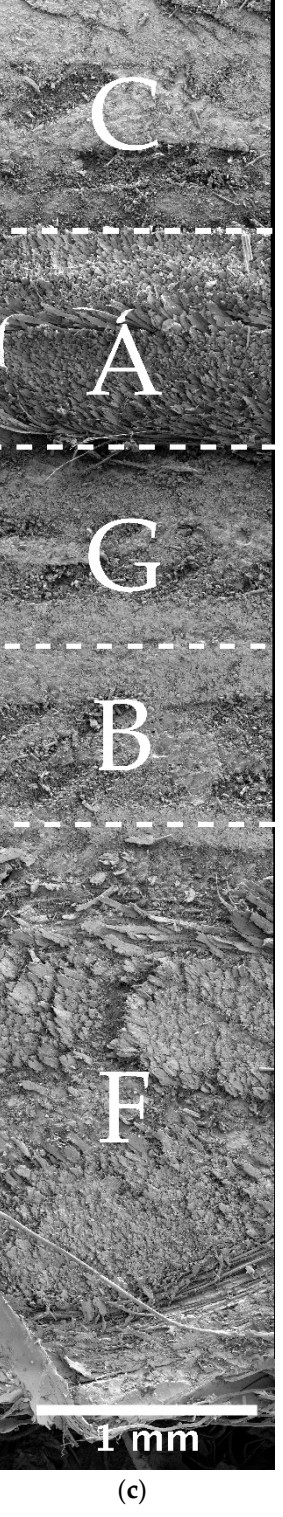

(**c**)

**Figure 1.** *Cont.*

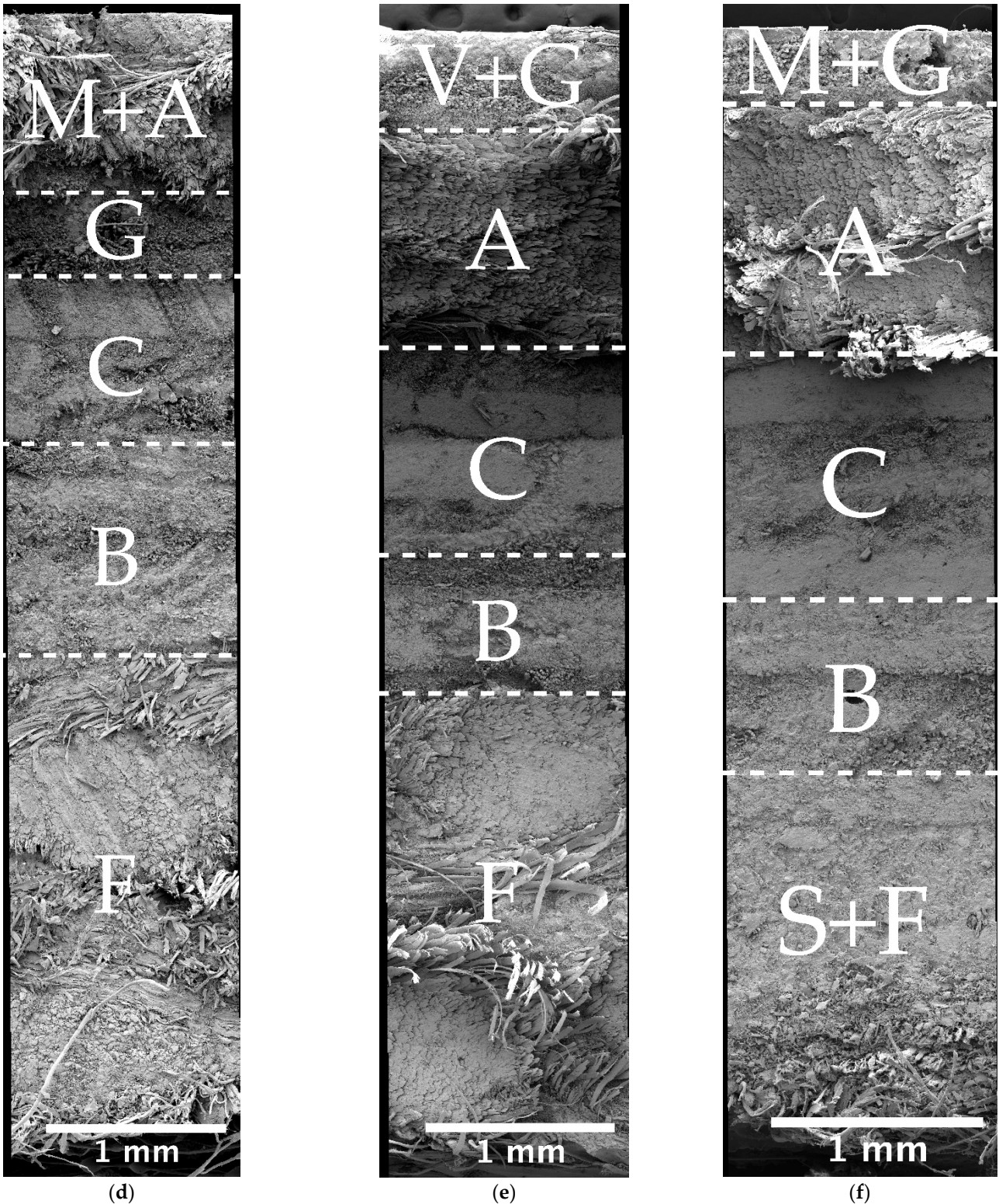

**Figure 1.** SEM images of cross-section of samples: GCABF (**a**), CGABF (**b**), CAGBF (**c**), MAGCBF (**d**), VGACBF (**e**), MGACBSF (**f**), magnification ×100.

## 3.2. Mechanical Property Evaluation

The characterization of mechanical properties of epoxy composites reinforced by fabrics and powder fillers were carried out and the results obtained from the experimentation are given in Figures 2–5.

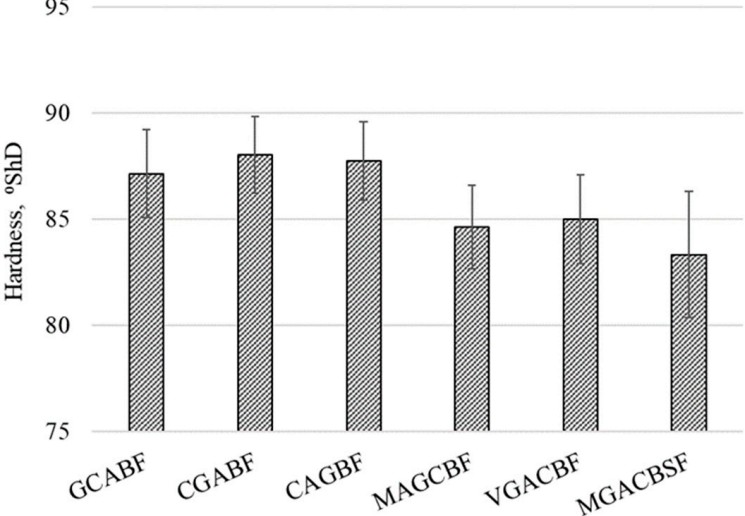

**Figure 2.** Hardness of EP-laminates.

Figure 2 presents the comparison of hardness results of investigated hybrid composites. The highest values were reached for CGABF (88.0° ShD) and CAGBF (87.7° ShD), and was higher by 5% than the lowest result noted to MGACBS. It was observed that the highest hardness was obtained for composites, whose top layer was made of carbon fabric. Moreover, replacing the layers with powder fillers resulted in a decrease in the parameter, which was further reduced along with an increase in replaced fabrics. The loads applied through the indenter led to compressive force increases, which result in fibers and particles touching each other and causing resistance [20]. It can be concluded that higher resistance is offered by composites with a higher amount of fabrics.

In the literature, we can find examples confirming that hybridization enhanced the flexural strength and modulus of composites [21,22]. Based on the flexural test, the flexural strength ($\sigma_f$) as well as Young's modulus (Et) values were determined and summarized in Figure 3.

The MAGCBF and VGACBF showed the highest flexural strength, and the values obtained by them amounted to approx. 200 MPa. Among the remaining composites, the strength above 150 MPa was recorded for MGACBSF and MAGCBF, and the lowest value was determined for CGABF. The analysis of the results and compositions allowed observing that the order of the fabrics had a greater influence on the results than the presence of powder fillers. The laminates are both expanded and compressed during the bending, whereas a small amount of filler does not influence the composites' compression properties considerably. The results are compatible with those presented by Oleksy [23], who observed that the addition of a few percentages of modified bentonites to Kevlar/epoxy composites did not significantly affect the bending stress of laminates. Moreover, the highest obtained values were recorded for composites, in the case of which, under the mineral powder filler, two successive layers were aramid and glass. Sarasini et al. [22] indicated that placing the G-fiber at the specimen skin and F-fiber in the core led to improvements in flexural strength. The influence of the order of reinforcement during examining hybrid composites made of carbon, flax, and basalt fibers was also observed by Abd El-Baky et al. [21]. This is due to the fact that the flexural strength and stiffness are controlled by the outer layers [22,24,25]. From Figure 3b, it is evident that the values above 8 GPa were obtained for the composites VGACBF, CAGBF, and MAGCBF. On the other hand, the lowest one, similar to hardness investigation, was obtained for the composite MGACBFS, containing the lowest amount of fabrics.

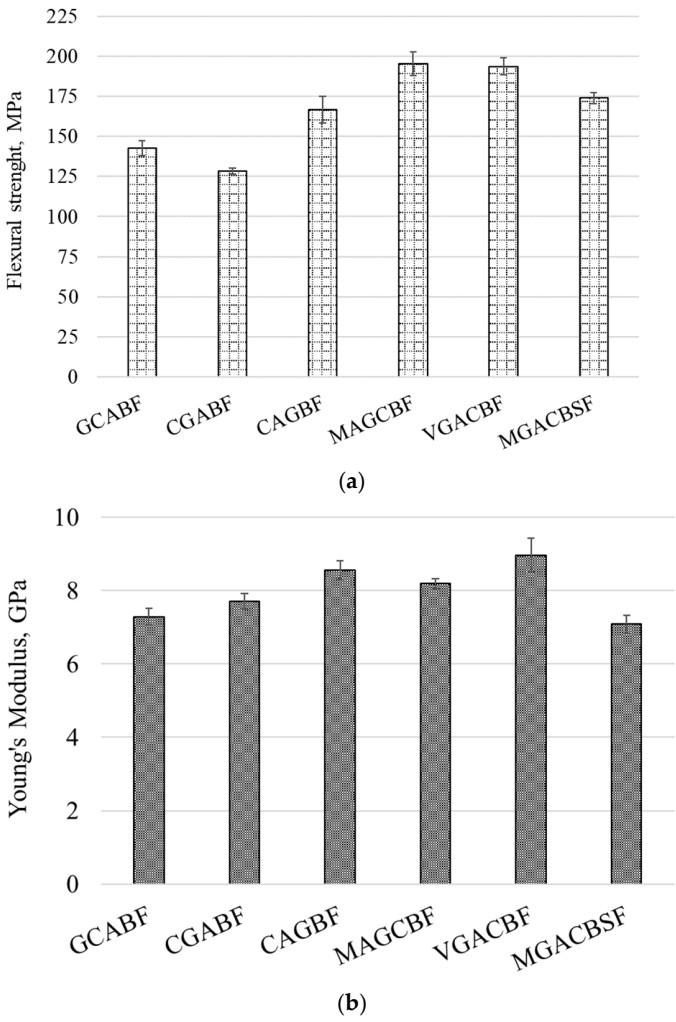

(a)

(b)

**Figure 3.** Flexural strength (**a**) and Young's Modulus (**b**) results of EP-composites obtained from the flexural test.

The mechanical properties' evaluation was completed by comparing the mechanical properties of composites in terms of puncture impact behavior.

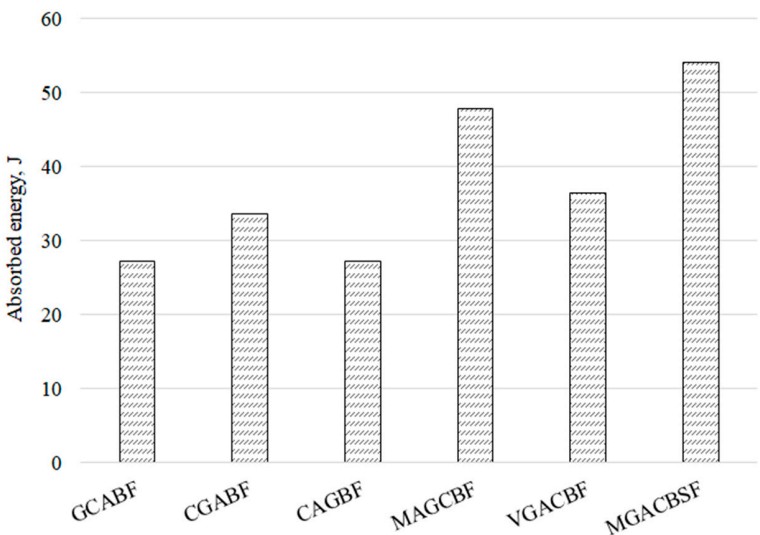

**Figure 4.** Absorbed energy of EP-laminates.

The effect of aramid, carbon, basalt, glass and flax fabrics of similar grammage, as well as powder fillers, on the energy absorbed by the composites, is illustrated in the data of Figure 4. It can be observed that the maximum value of absorbed energy is shown by MGACBFS as well as MAGCBS, both containing glass spheres in an outer layer. In turn, GCABF and CAGBF have the largest brittleness from all investigated composites. The results evident that a higher resistance is offered by composites with powder fillers.

Figure 5 shows impacted laminates which depict various damage modes under the low-velocity impact. A visual observation illustrated damage patterns, including delamination around the impact point, fiber breakage and debonding. All this failure mode is similar to common damage modes in composite laminates and depends on the impact energy and impactor shapes. The greatest visual delamination around the impact point, reaching even 10 mm, was observed in the GCABF. In comparison with the greatest delamination area, it is seen that MAGCBF shows minimal borders of the visual delamination around the impact point (approx. 2 mm).

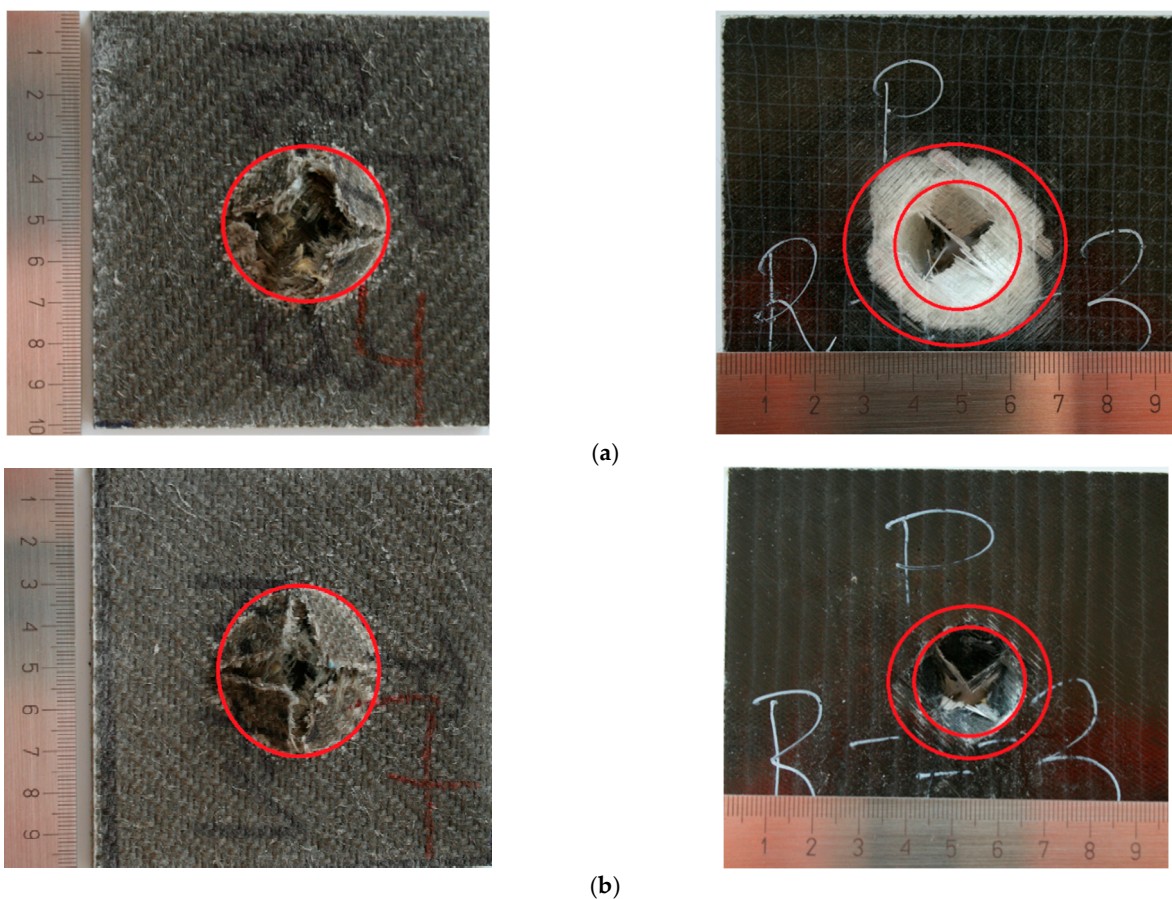

(a)

(b)

**Figure 5.** *Cont.*

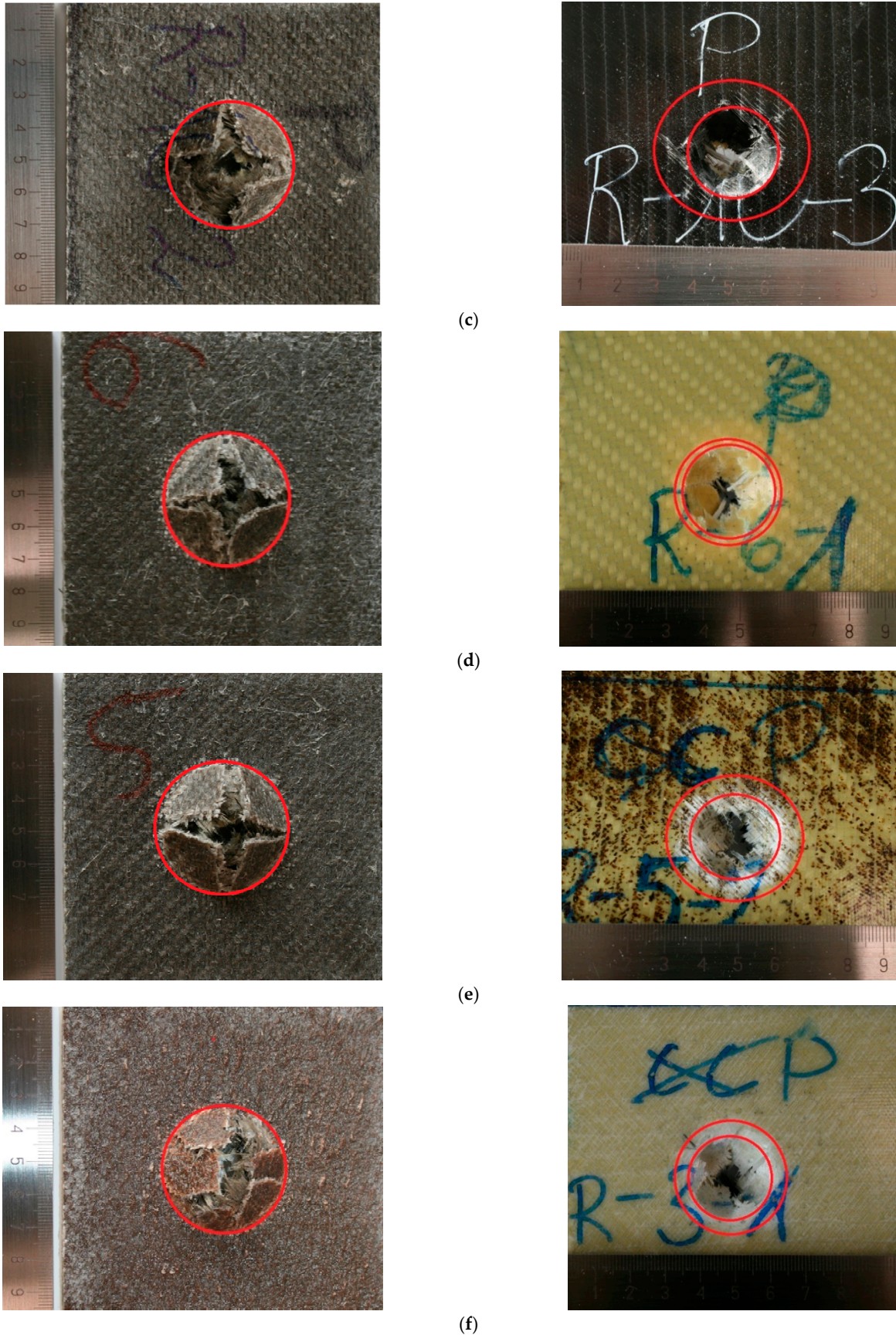

**Figure 5.** Photographs of impacted laminates GCABF (**a**), CGABF (**b**), CAGBF (**c**), MAGCBF (**d**), VGACBF (**e**), MGACBSF (**f**) with various damage modes.

### 3.3. Thermal Stability

The thermal stability of hybrid composites was interpreted through TG and DTG curves, as illustrated in Figure 6. Table 2 details the various temperatures obtained from thermograms, such as the initial temperature of the decomposition, maximum decomposition temperature with the degradation rate and residue in 900 °C.

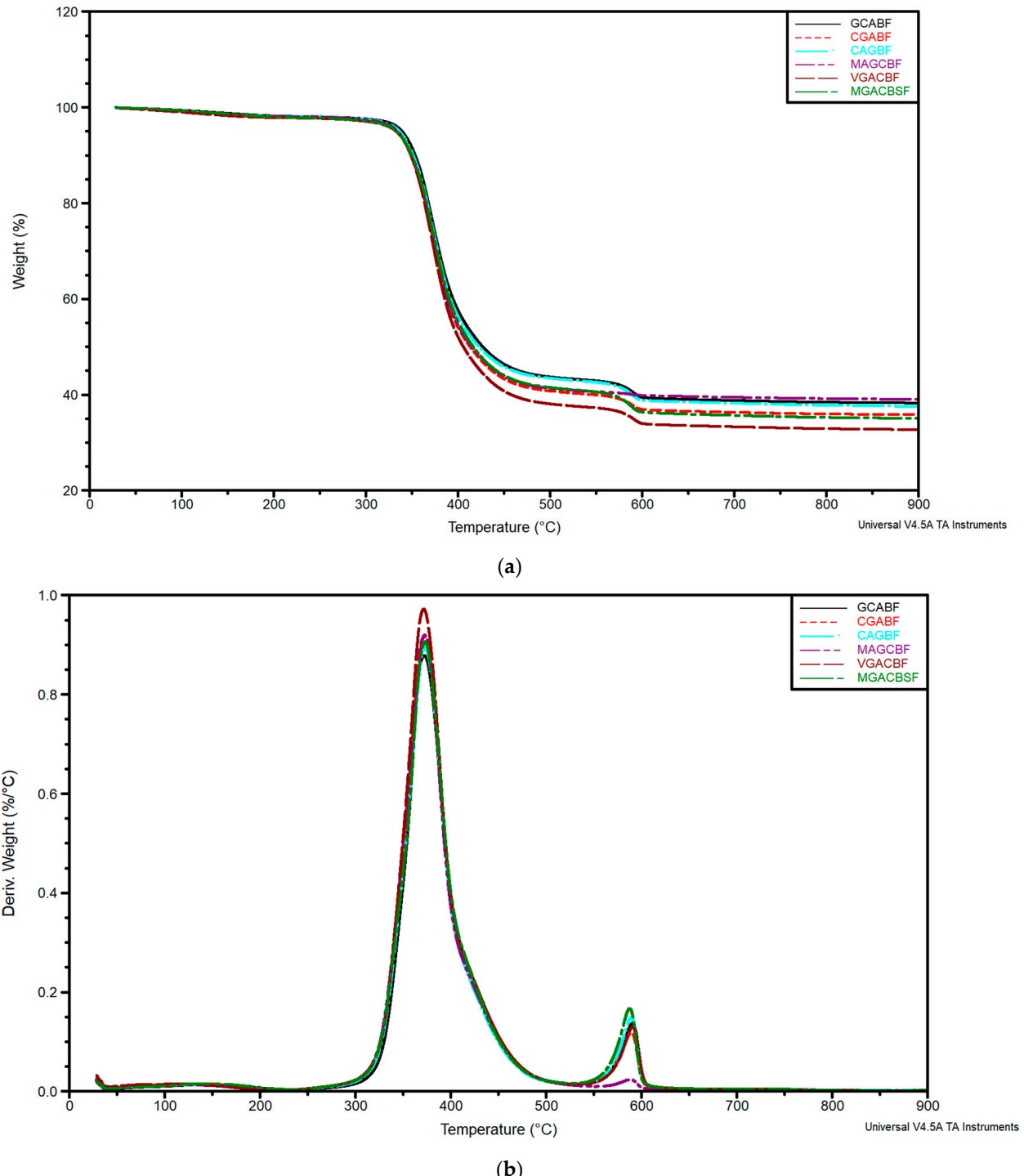

**Figure 6.** TG (**a**) and DTG (**b**) curves of EP-composites obtained from the thermos gravimetric analysis.

**Table 2.** TG and DTG data of unmodified UP and resin with APP and LHP.

| Samples | $T_{5\%}$, °C | DTG1, °C; %/°C | DTG2, °C; %/°C | DTG3, °C; %/°C | Residue in 900 °C, % |
|---------|------|------|------|------|------|
| GCABF | 339 | 146; 0.01 | 372; 0.88 | 590; 0.14 | 38.3 |
| CGABF | 333 | 137; 0.02 | 372; 0.92 | 589; 0.12 | 25.8 |
| CAGBF | 336 | 135; 0.01 | 373; 0.89 | 589; 0.15 | 37.6 |
| MAGCBF | 334 | 138; 0.01 | 372; 0.92 | 588; 0.02 | 39.0 |
| VGACBF | 333 | 124; 0.02 | 371; 0.97 | 591; 0.13 | 32.7 |
| MGACBSF | 333 | 143; 0.02 | 373; 0.91 | 587; 0.17 | 35.1 |

The course of the TG and DTG curves to the temperature of approx. 400 °C was similar for all laminates, which results from the use of the same fabrics, being the dominant component in the investigated composites. The small sample size meant that the increasing temperature affected the entire volume of the sample, and the varying order of the layer did not affect the results so much. The 5% weight loss temperature (T5%), corresponding to the decomposition's onset temperature, was similar and ranged from 333 to 339 °C. As shown in Table 2, slightly lower values were obtained mainly for composites with powder fillers (VGACBF, MGACBSF), nonetheless, it was not a rule. It can be concluded that the investigated composites which are thermally stable are capable enough to be used in workable applications until 330 °C [26]. The laminates had three steps of degradation at 124–146 °C (DTG1), 371–373 °C (DTG2), and 587–591 °C (DTG3), respectively. The first peak corresponds to the elimination of water and low molecular weight compounds. The most intense decomposition occurred at 372 °C, which correlates with the degradation of the epoxy resin aromatic group's and the curing agent's aliphatic amine [27], but also may be related to degradation of hemicellulose from organic fillers. At this stage, the decomposition rate was the highest for all materials and reached almost 1%/°C. The last stage corresponds to other fabrics' decomposition, such as aramid, suggested by a lower degradation rate for MAGCBF, in which glass spheres replaced the A fabric. The similar composition of laminates led to obtaining a similar residue yield, which reached from 33 to 39% and was lower only in the case of CGABF (26%).

### 3.4. Fire Behavior

The cone calorimetry is well-known, an effective bench-scale tool for investigating the polymer and its composites' burning behavior. This technique provides data, which may correlate with the results received from full-scale experiments [28]. The CC experiments' results are presented in Table 3 and Figure 7.

**Table 3.** Cone calorimeter results of EP-laminates.

| Samples | TTI, s | pHRR, kW/m² | MARHE, kW/m² | THR, MJ/m² | SEA, m²/kg | TSR, m²/m² |
|---------|--------|------|------|------|------|------|
| GCABF | 127 (37 [a]) | 462 (96) | 216 (41) | 139 (9) | 550 (21) | 3378 (29) |
| CGABF | 175 (91) | 637 (135) | 204 (16) | 121 (5) | 508 (23) | 3053 (29) |
| CAGBF | 96 (11) | 558 (75) | 232 (1) | 153 (3) | 519 (14) | 3347 (134) |
| MAGCBF | 142 (1) | 706 (104) | 257 (2) | 136 (8) | 532 (14) | 3172 (168) |
| VGACBF | 121 (16) | 454 (122) | 190 (14) | 125 (3) | 563 (44) | 3210 (240) |
| MGACBSF | 94 (4) | 571 (223) | 214 (26) | 118 (8) | 556 (5) | 2931 (39) |

[a] The values in parentheses are the standard deviations.

The heat release rate versus time curves are shown in Figure 7. As can be seen, all the curves contain two clearly visible peaks and a plateau-like behavior between them. The most composites' HRR curves present a peak heat release rate (pHRR) at the end of the burning before the HRR suddenly decreases. The opposite trend was observed only in the case of CABGF, for which the first maximum yield is the pHRR. Moreover, the intensity of

burning depended significantly on the type and the order of fabrics in composites. The detailed data are shown in a table.

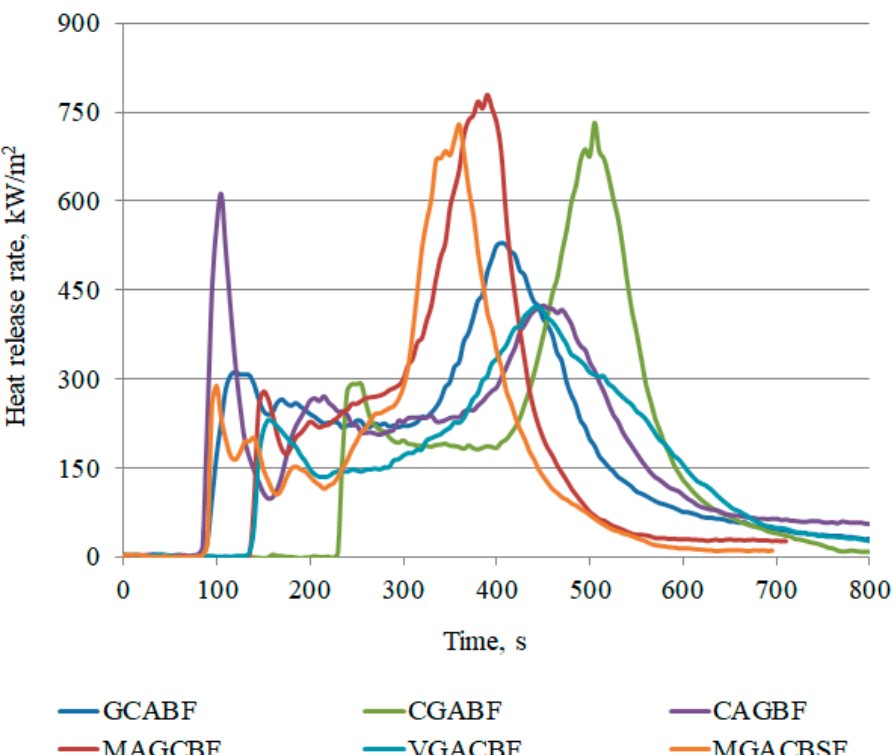

**Figure 7.** Representative curves of heat release rate of the EP-based composites.

The time to ignition (TTI), which defines the time required for the ignition and maintenance of the flame over the sample surface [29], achieved 94 to 142 s, and only for CGAB was significantly higher (172 s). However, the standard deviation in the case of these composites was also the highest. It should be emphasized that no significant trend associated with the sequence of fabrics was observed. On the other hand, the heat release rate, as the most evaluative factor among the fire properties, was ranging from 454 to 706 kW/m$^2$, and the lowest values were obtained for composites with glass fabrics (GCABF) or vermiculite powder with glass fabric (VGACBF) as the outer layer. Since the amount of the epoxy resin, which is the most flammable component in investigated materials, was similar, it confirms the crucial influence of the type, form, and sequence of fabrics. For example, the replace of carbon fabrics with glass ones in CGCBF and GACBF composites decreased the pHRR by 38%, while glass spheres by vermiculite in MGACBSF and VGACBF decreased by 26%. This may be connected with the release of water, physisorbed in the vermiculite, that occurred due to the external heat flux impact and led to the exfoliation of V at an elevated temperature. Vermiculite, with a structural formula (Mg,Fe,Al)$_3$(Al,Si)$_4$O$_{10}$(OH)$_2$·4H$_2$O), is known for its flame retardant properties, related to the high resistance to heat and ability to water absorption [27,30,31]. Similarly, the lowest values of the maximum average rate of heat emission (MARHE), regarded as a parameter to estimate the hazard of fire spread [32], were obtained for VGACBF. In turn, the highest pHRR and MARHE were obtained for the composite, in which the outer layer contains mainly glass spheres and aramid fibers.

No evident trend linking the form or sequence of the fillers with total heat release (THR) values can be observed. However, lower THR values were obtained for MGACBSF, VGACBF, and CGABF, in which under carbon fabrics or glass spheres and vermiculite are layers of glass and aramid. In turn, putting aramid fabrics as the second layer (CABGF, MAGCBF) or replacing carbon fabrics with glass ones (GCACBF) led to increasing in THR. Unusually, a lower THR, considered a measurement for the fire load of material, suggests incomplete combustion due to the reduced combustion efficiency or formation of a char [33].

Figure 8 presents the photographs of composites after cone calorimeter measurements. As can be seen, all the residues consist mostly of reinforcement, and the charred residue on the surface of the samples is barely visible (Figure 8b). Additionally, shimmering powder particles can be observed in the case of composites with vermiculite (Figure 8e).

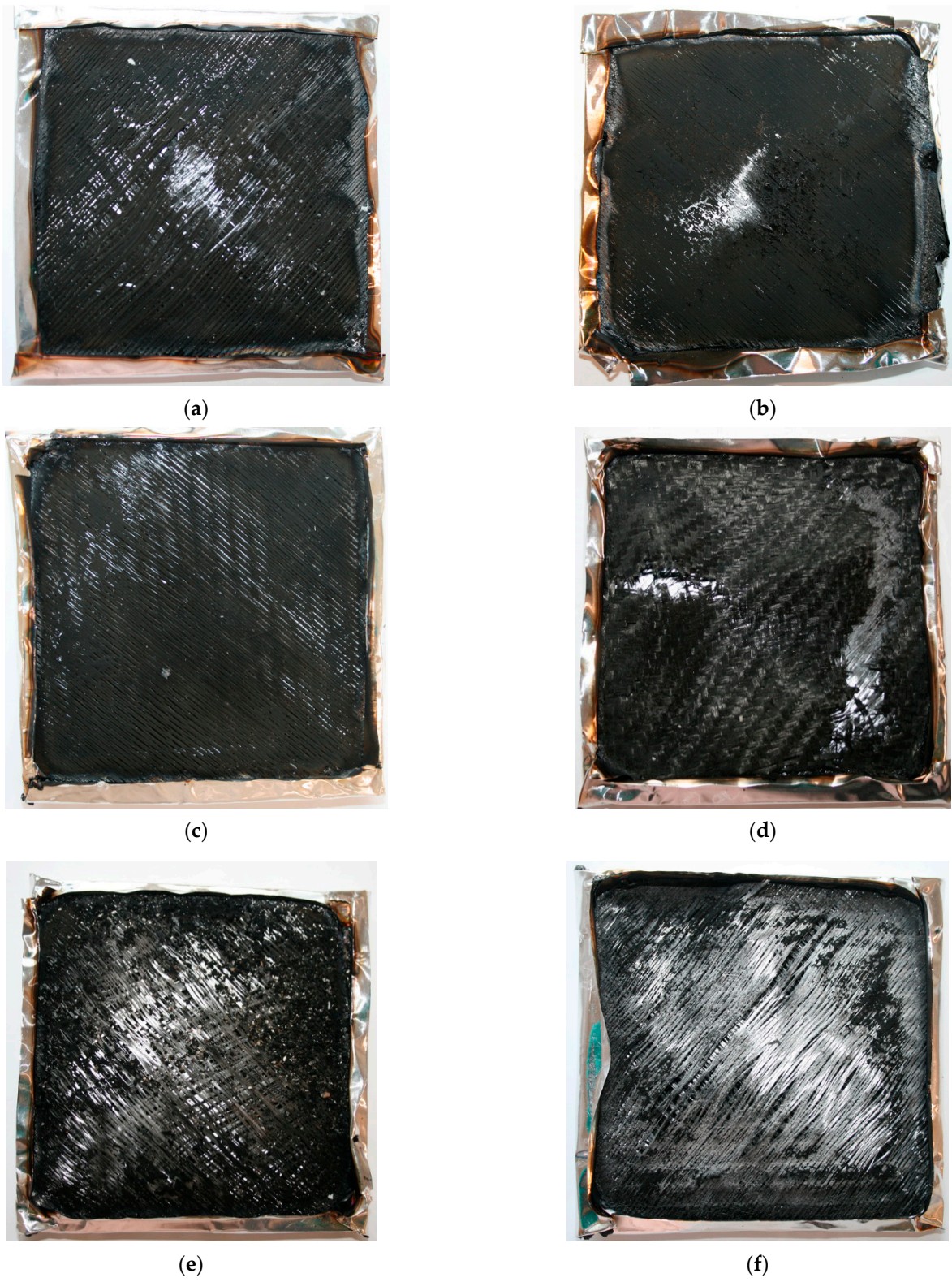

(a)

(b)

(c)

(d)

(e)

(f)

**Figure 8.** Photographs of GCABF (**a**), CGABF (**b**), CAGBF (**c**), MAGCBF (**d**), VGACBF (**e**), MGACBSF (**f**) after cone calorimetry tests.

In order to check the effect of the CC test on the appearance and quality of fabrics in the laminates, they were subjected to the microstructure analysis using SEM. Figure 9 shows the original appearance of individual fabrics (left column) and subsequent layers of a composite VGACBF after the CC test (right column). As can be seen, the most damaged is the G fabric, which was also the most outer layer of laminates. Below it there is a less damaged A with fragments of glass spheres attached to the fibers. The next layer was the least damaged carbon fabric, the fibers of which look almost identical to the original ones. Then, the basalt fabric, interestingly more damaged than the carbon one, as well as linen, whose fibers are still stuck together by the polymer matrix. Linen fibers bonded together by epoxy resin and juxtaposed with the original appearance of the fabric at a lower magnification are shown in Figure 10. This means that the layers of fabrics above prevented the resin from burning out completely. In turn, Figure 11 shows the appearance of aramid fibers, and using a higher magnification, allowed confirming the presence of agglomerates of glass spheres and denying their complete fragmentation into a powder.

Before CC test        After CC test

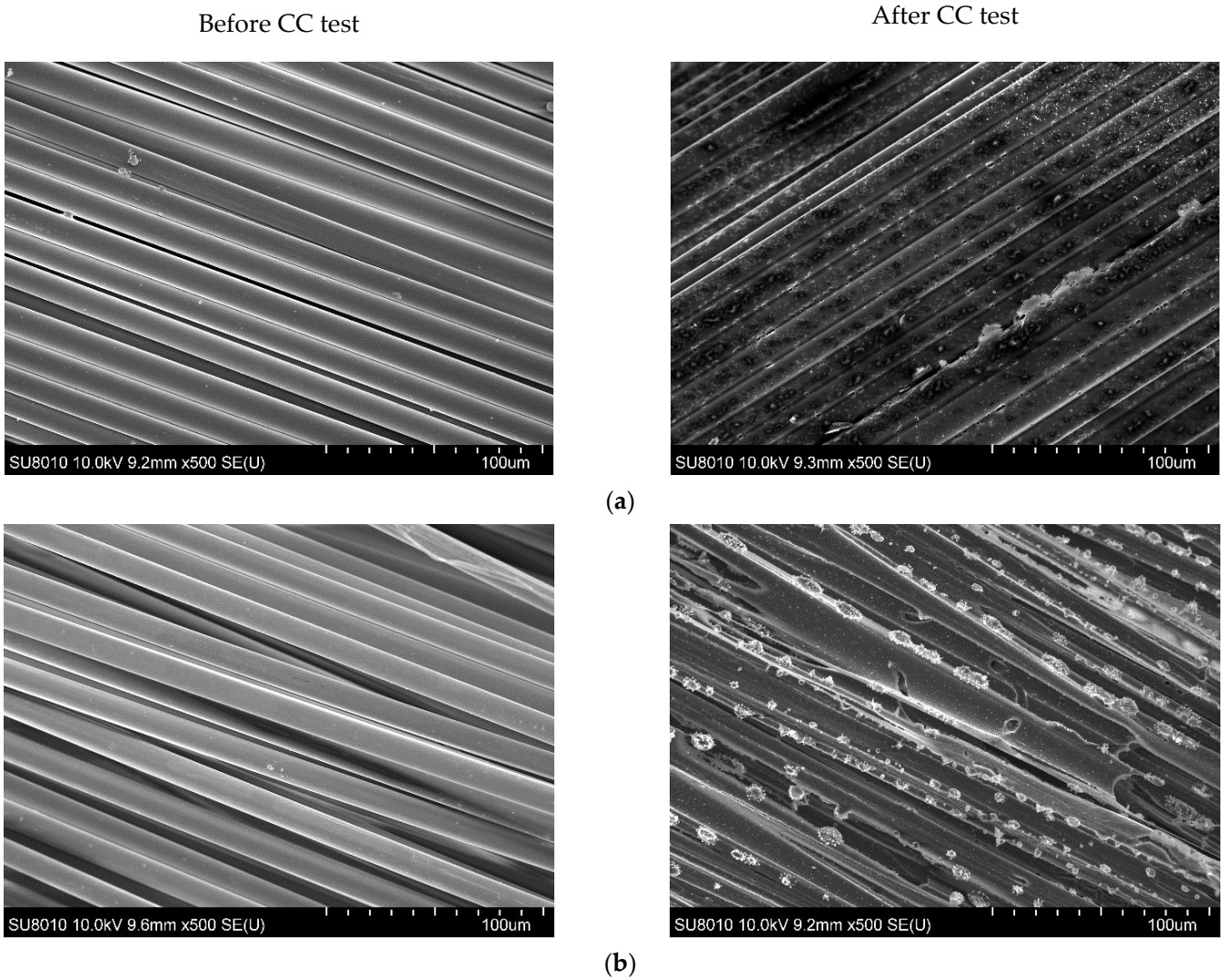

(**a**)

(**b**)

**Figure 9.** *Cont.*

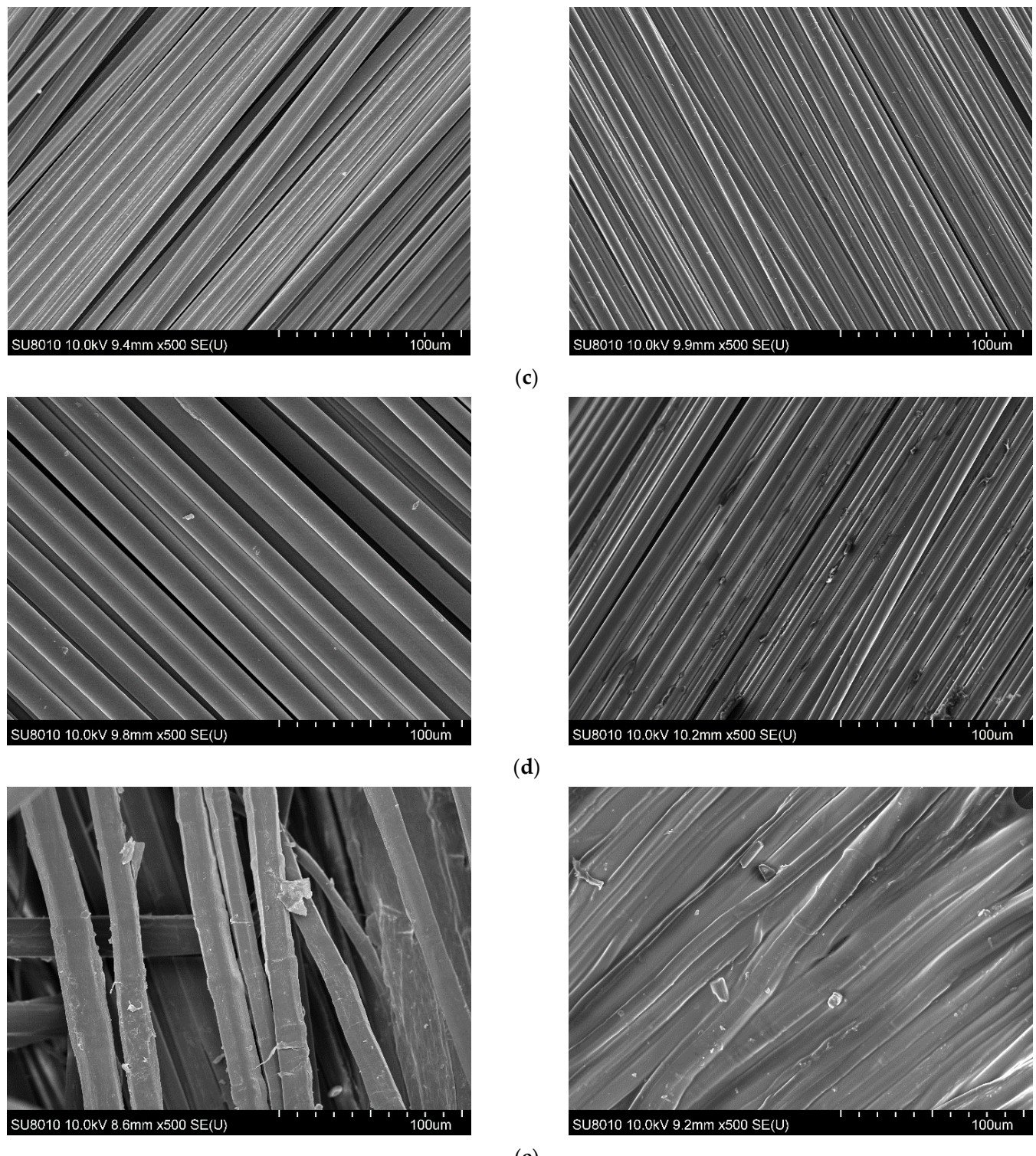

**Figure 9.** SEM images of G (**a**), A (**b**), C (**c**), B (**d**), and F (**e**) fabrics before CC tests (left column) and VGACBF composites after CC tests (right column).

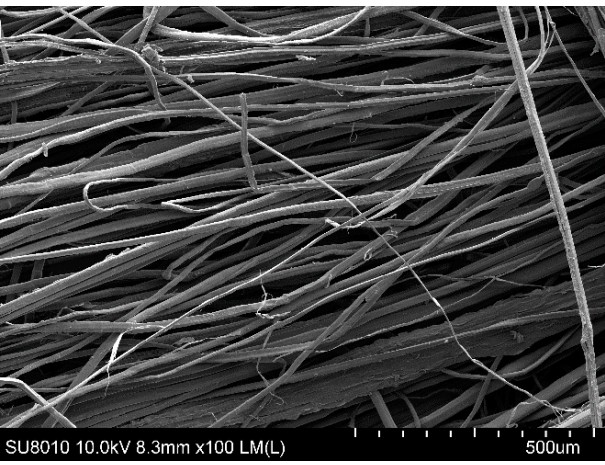
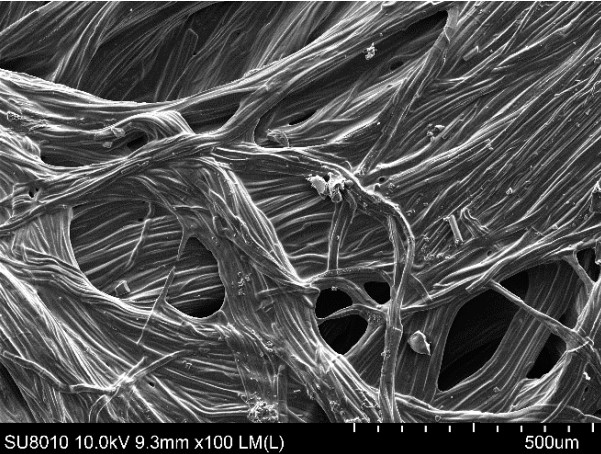

**Figure 10.** SEM images of F fabrics before CC tests (left column) and bottom of VGACBF composites after CC tests (right column).

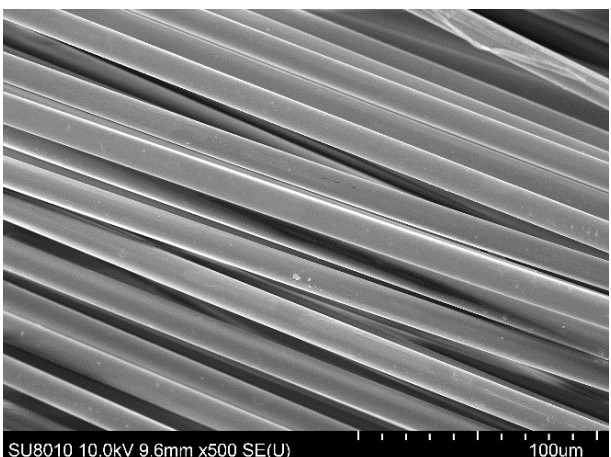
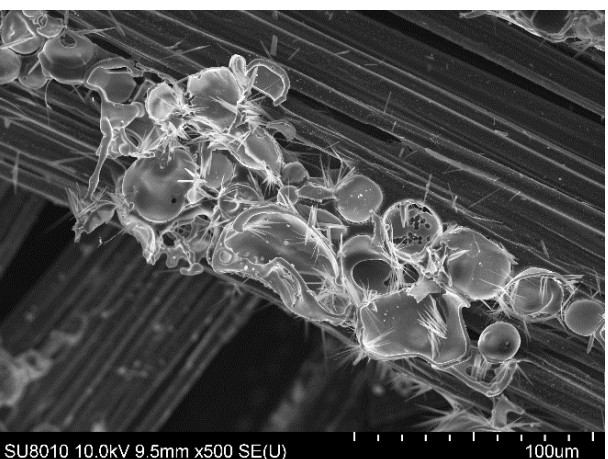

**Figure 11.** SEM images of A fabrics before CC tests (left column) and top of MGACBSF composites after CC tests (right column).

The smoke emission was assessed based on the specific extinction area (SEA) and total smoke release (TSR). The SEA, which corresponds to the light absorption by the particles of smoke formed during the burning of 1 kg of the material and indirectly informs about visibility during the fire [34], ranged from 508 to 563 $m^2$/kg. The lowest values were obtained for composites with carbon fabric as the most outer layer (Table 3). In turn, the lowest TSR was recorded for MGACBSF (2931 $m^2$/$m^2$), but also for CGABF (3053 $m^2$/$m^2$), for which the greatest delay in emissions was also observed (Figure 12). This suggests that the powder fillers' presence may have a positive effect on the reduction of the amount of smoke released. The reduction of TSR as a result of introducing ground hazelnut shell into epoxy resin was described in our earlier works [35,36].

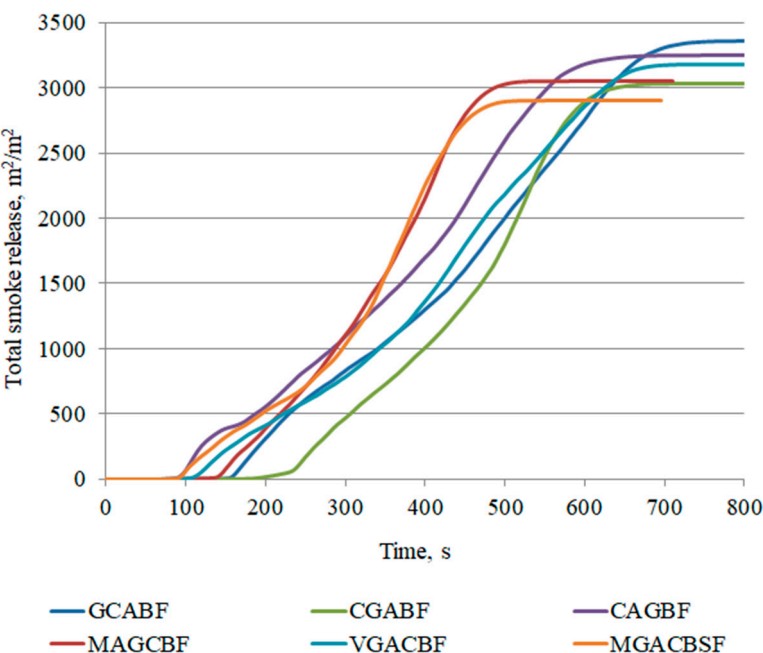

**Figure 12.** Representative curves of total smoke release of the EP-based composites.

## 4. Conclusions

Aramid, carbon, basalt, glass, and flax fabrics, as well as powder fillers such as the glass spheres, vermiculite, and ground hazelnut shells, were employed to manufacture the hybrid composites.

Microstructure observations confirmed that the use of a low-viscosity epoxy resin had a positive effect on the saturation of fabrics. However, SEM images showed that agglomerates also formed in the case of the mineral fillers as well as pores. The analysis of results of the flexural test showed that the order of the fabrics had a greater impact on the results than their replacement by powder fillers. In turn, replacing the layers with powder fillers resulted in a decrease in the hardness.

The similar composition of laminates resulted in comparable thermal stability composites. However, significant differences in their burning behavior were observed. Since the amount of resin, as the most flammable component, was the same, it confirms the impact of the type and form of fillers, as well as the sequence of fabrics. The lowest pHRR was obtained for composites with glass fabrics or vermiculite powder with glass fabrics as the outer layer. Moreover, a positive effect on the reduction of the amount of smoke released resulting from the powder fillers' presence was noted.

**Author Contributions:** Conceptualization, K.S. and M.K.; methodology, K.S., A.K., E.S., M.C., K.M., P.K. and K.K.; software, K.S., P.C., E.S., A.K. and P.K.; formal analysis, K.S., P.C. and A.K.; investigation, K.S., M.K., P.C., E.S., A.K., M.C., K.M., P.K. and M.G.; resources, K.S., K.M., U.C. and K.K.; writing—original draft preparation, K.S., P.C., A.K. and K.M.; visualization, K.S. and A.K.; supervision, U.C. and K.K.; project administration, K.S.; funding acquisition, K.S. All authors have read and agreed to the published version of the manuscript.

**Funding:** This paper has been based on the results of a research task carried out within the scope of the fifth stage of the National Programme "Improvement of safety and working conditions" partly supported in 2020–2022—within the scope of research and development—by the Ministry of Education and Science/National Centre for Research and Development. The Central Institute for Labour Protection—National Research Institute is the Programme's main coordinator.

**Institutional Review Board Statement:** Not applicable.

**Informed Consent Statement:** Not applicable.

**Data Availability Statement:** The data presented in this study are available on request from the corresponding author.

**Acknowledgments:** The research was realized with equipment allocated to the Central Institute for Labour Protection—National Research Institute, Warsaw University of Technology, Latvian State Institute of Wood Chemistry and Riga Technical University.

**Conflicts of Interest:** The authors declare no conflict of interest.

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
