# Peer review of "Experimental Investigation of the Mechanical Properties and Fire Behavior of Epoxy Composites Reinforced by Fabrics and Powder Fillers"

_processes, doi:10.3390/pr9050738_

Round 1

Reviewer 1 Report

The inclusion of flax fabric and hazelnut shell is in contrast to mechanical and flame protection. The authors should provide sufficient motivation to include such material alongside high performance fabrics.

The SEM images are without resolution, scale and magnification. The authors must include the details.

Author Response

Dear Reviewer,

We highly appreciate all the comments and find them very useful. We agree with the recommendations that the manuscript should be improved, so efforts have been made to correct the article according to the comments. Below we enclose the replies to the comments and recommendations made by the Reviewer. All of the changes in the text are highlighted using colour.

  1. The inclusion of flax fabric and hazelnut shell is in contrast to mechanical and flame protection. The authors should provide sufficient motivation to include such material alongside high performance fabrics. 

We thank the Reviewer for very valuable advice and agree that the main idea of our research was not properly presented in the article. Bearing in mind the Reviewer’s remark we changed the last paragraph in the introduction part by adding a few sentences of explanation.

  1. The SEM images are without resolution, scale and magnification. The authors must include the details.

We thank the Reviewer for the remarks. Respective changes have been taken into consideration.

In order to stitch the images, default scale bars have been removed. This is why observation conditions are listed in the Materials and Methods section of the article:

 “Observations were conducted in the secondary electron (SE) mode, at an accelerating voltage of 10 kV, at magnification x100 and the biggest possible working distance, usually WD>30 mm, to maximize field depth and minimize image distortion.”

Addition information regarding the image’s resolution has been added:

“Final resolution of the stitched images was around 2600x13000 pixels, which corresponds to horizontal field of view 1.3 mm and vertical field of view > 6 mm.”

Moreover, a scale bar has been added to the images in Figure 1 with the magnification information in the figure’s caption.

We have also corrected quite a number of other minor errors which we noticed while working on the text, and we believe that our manuscript in the present form can be published in the Journal.

Yours faithfully,

Kamila Salasinska,

Paweł Kozikowski

Reviewer 2 Report

Dear Authors!

I have reviewed your manuscript "Experimental investigation on the mechanical properties and fire behavior of epoxy composites reinforced by fabrics and powder fillers" and found generally acceptable for publication in MDPI "Processes" after minor revision. Please, find below my comments.

I. Typos and phrasing:

  • Title: "Experimental investigation on the mechanical properties and fire ..." May be ".... of mechanical properties ..."?
  • lines 23-24 "The mechanical properties' evaluation was related to the microstructure and completed with thermogravimetric analysis (TGA)."  - may be "The mechanical properties were evaluated to correlate with  the microstructure and to consider together with thermogravimetric analysis (TGA) data"
  • line 37  "A promising " 
  • line 53 MPA -> MPa
  • line 99 and 101 - please, check BIAX 400 g. Is it the same tradename for both glass and carbon fabrics?
  • line 108. Please, more details for Vermeko.
  • line 315. "in which the most outer layer consists of glass spheres and aramid fibres. " -may be "in which outer layer contains mainly glass spheres and aramid fibres." ? 

II. Requests for more information and/or recommendations:

  • line 126. Tools for cutting. Knifes? Milling? Laser cutter?
  • lines 134-135. Please, find a chance to mark layers corresponding to different fabrics in Figure 1. 
  • Also in Figure 1 Scale bar (we know, that thickness is 6 mm, however, it is common practice to give scale bar).
  • Figure 1 - SE or BSE? Voltage?
  • What is depicted - edge of as-cured sample or cross-section after cutting?
  • line 213 "During the bending, the laminates are mainly compressed ..." Please, rephrase. During the bending a plate is both expanded and compressed as continuous mechanics dictates. 
  • line 235. "In turn, GCABF and CAGBF have the largest brittleness of the polymer matrix. " Please, rephrase. Epoxy matrix shouldn't be specifically brittle in these composites.
  • lines 246-248. Any chance to quantify the sizes of delamination/punch area?
  • line 335. What was the technique to open layers in VGACBF after CC? Mechanical delamination? 

III. Please, highlight the main idea of your research in Introduction. Combination of different fabrics should improve performance? Reduce costs? Renewable raw materials?  

Regards ...

Author Response

Dear Reviewer,

We highly appreciate all the comments and find them very useful. We agree with the recommendations that the manuscript should be improved, so efforts have been made to correct the article according to the comments. Below we enclose the replies to the comments and recommendations made by the Reviewer. All of the changes in the text are highlighted using colour.

  1. Title: "Experimental investigation on the mechanical properties and fire ..." May be ".... of mechanical properties ..."? 

We thank the Reviewer for the remark. The title has been changed accordingly.

  1. lines 23-24 "The mechanical properties' evaluation was related to the microstructure and completed with thermogravimetric analysis (TGA)." - may be "The mechanical properties were evaluated to correlate with  the microstructure and to consider together with thermogravimetric analysis (TGA) data".

Thanking the Reviewer for the remarks we would like to say that the suggested changes have been made.

  1. line 37 "A promising "

The suitable corrections according to the Reviewer’s comment have been introduced.

  1. line 53 MPA -> MPa

The authors wish to thank the Reviewer for drawing attention to the errors that were made. Identified deficiencies have been corrected in the text.

  1. line 99 and 101 - please, check BIAX 400 g. Is it the same tradename for both glass and carbon fabrics?

We thank the Reviewer for comment. We checked it on the producer website and carefully improved the names, as well as grammage, of fabrics in the manuscript. The previous information was obtained from the supplier and was not correct.

  1. line 108. Please, more details for Vermeko.

Thanking the Reviewer for calling our attention to the issue the authors would like to say that every effort has been made to improve the description.

  1. line 315. "in which the most outer layer consists of glass spheres and aramid fibres. " -may be "in which outer layer contains mainly glass spheres and aramid fibres." ?

We thank the Reviewer for the suggestion. The suitable corrections in the sentence according to the comment have been introduced.

  1. line 126. Tools for cutting. Knifes? Milling? Laser cutter?

The proper information was added to the Methods section.

  1. lines 134-135. Please, find a chance to mark layers corresponding to different fabrics in Figure 1.

Fabrics have been appropriately marked in Figure 1. Layers have been divided by dotted lines and marked with the letter corresponding to the layers.

  1. Also in Figure 1 Scale bar (we know, that thickness is 6 mm, however, it is common practice to give scale bar).

Scale bar has been added to each image.

  1. Figure 1 - SE or BSE? Voltage?

Observation conditions are listed in the Materials and Methods section of the article. Information about the electron mode has added:

Line 132: “Observations were conducted in the secondary electron (SE) mode, at an accelerating voltage of 10 kV, at magnification x100 and the biggest possible working distance, usually WD>30 mm, to maximize field depth and minimize image distortion”. 

  1. What is depicted - edge of as-cured sample or cross-section after cutting?

We thank the Reviewer for precious advice and agree that the description of Fig. 1 should be more informative.  

  1. line 213 "During the bending, the laminates are mainly compressed ..." Please, rephrase. During the bending a plate is both expanded and compressed as continuous mechanics dictates.

We thank the Reviewer for the valuable comment. We agree with the recommendation and carefully improved that inaccuracy in the manuscript.

  1. line 235. "In turn, GCABF and CAGBF have the largest brittleness of the polymer matrix. " Please, rephrase. Epoxy matrix shouldn't be specifically brittle in these composites.

We appreciate the Reviewer’s suggestion and agree that the information may cause a lack of clarity and that’s a way it has been improved.

  1. lines 246-248. Any chance to quantify the sizes of delamination/punch area?

We thank the Reviewer for the comment and agree that the information about the sizes of the delamination area should be included in the description of the results.  

  1. line 335. What was the technique to open layers in VGACBF after CC? Mechanical delamination?

Each layer after CC was brittle and easily delaminated by tweezers. According to the Reviewer’s comment, suitable corrections have been introduced in the Materials and Methods section.

  1. Please, highlight the main idea of your research in Introduction. Combination of different fabrics should improve performance? Reduce costs? Renewable raw materials?

We thank the Reviewer for very valuable advice and agree that the main idea of our research was not properly presented in the article. Bearing in mind the Reviewer’s remark we changed the last paragraph in the introduction part by adding a few sentences of explanation.

We have also corrected quite a number of other minor errors which we noticed while working on the text, and we believe that our manuscript in the present form can be published in the Journal.

Yours faithfully,

Kamila Salasinska,

Paweł Kozikowski
